# Effects of Gas Production Recording System and Pig Fecal Inoculum Volume on Kinetics and Variation of In Vitro Fermentation using Corn Distiller’s Dried Grains with Solubles and Soybean Hulls

**DOI:** 10.3390/ani9100773

**Published:** 2019-10-09

**Authors:** Jae-Cheol Jang, Zhikai Zeng, Gerald C. Shurson, Pedro E. Urriola

**Affiliations:** 1Department of Animal Science, University of Minnesota, St. Paul, MN 55108, USA; jang0046@umn.edu (J.-C.J.); shurs001@umn.edu (G.C.S.); 2Department of Veterinary Population Medicine, University of Minnesota, St. Paul, MN 55108, USA

**Keywords:** corn distillers dried grains with solubles, gas collection technique, in vitro, pig fecal inoculum, soybean hulls

## Abstract

**Simple Summary:**

Various in vitro methodologies have been developed and used to estimate the digestibility of feed ingredients, such as corn distillers dried grains with solubles (cDDGS) and soybean hulls (SBH) which contain high concentrations of dietary fiber. This study evaluated two in vitro gas production recording systems (manual vs. automated) and two initial fecal inoculum volumes (30 vs. 75 mL) on the parameters of in vitro fermentation of cDDGS and SBH. The results showed that the use of 75-mL inoculum volume with 0.5 g substrate tended to reduce the variation of measurements compared to the 30-mL inoculum volume with 0.2 g substrate regardless of the gas production recording system. These findings suggest that using larger inoculum volume with more substrate increases the precision of measurements. Furthermore, the automated system decreases labor for conducting the assay.

**Abstract:**

An experiment was conducted to investigate the effect of inoculum volume (IV), substrate quantity, and the use of a manual or automated gas production (GP) recording system for in vitro determinations of fermentation of corn distillers dried grains with solubles (cDDGS) and soybean hulls (SBH). A 2 × 2 × 2 factorial arrangement of treatments was used and included the factors of (1) ingredients (cDDGS or SBH), (2) inoculum volume and substrate quantity (IV30 = 0.2 g substrate + 30 mL inoculum or IV75 = 0.5 g substrate + 75 mL inoculum), and (3) GP recording system (MRS = manual recording system or ARS = automated recording system). Feed ingredient samples were pre-treated with pepsin and pancreatin, and the hydrolyzed residues were subsequently incubated with fresh pig feces in a buffered mineral solution. The GP recording was monitored for 72 h, and the kinetics were estimated by fitting data using an exponential model. Compared with SBH, cDDGS yielded less (*p* < 0.01) maximal gas production (*G_f_*), required more time (*p* < 0.02) to achieve half gas accumulation (*T*/2), and had less (*p* < 0.01) fractional rate of degradation (*µ*) and in vitro fermentability of dry matter (IVDMF). Using the ARS resulted in less IVDMF (*p* < 0.01) compared with MRS (79.0% vs. 81.2%, respectively). Interactions were observed between GP recording system and inoculum volume and substrate quantity for *Gf* (*p* < 0.04), *µ* (*p* < 0.01), and *T*/2 (*p* < 0.04) which implies that increasing inoculum volume and substrate quantity resulted in decreased *Gf* (332 mL/g from IV30 vs. 256 mL/g from IV75), *µ* (0.05 from IV30 vs. 0.04 from IV75), and *T*/2 (34 h for IV30 vs. 25 h for IV75) when recorded with ARS but not MRS. However, the recorded cumulative GP at 72 h was not influenced by the inoculum volume nor recording system. The precision of *G_f_* (as measured by the coefficient of variation of *G_f_*) tended to increase for IV30 compared with IV75 (*p* < 0.10), indicating that using larger inoculum volume and substrate quantity (IV75) reduced within batch variation in GP kinetics. Consequently, both systems showed comparable results in GP kinetics, but considering convenience and achievement of consistency, 75 mL of inoculum volume with 0.5 g substrate is recommended for ARS.

## 1. Introduction

The use of increasing amounts of dietary fiber (DF) in swine feeding programs contributes to various environmental [1], animal well-being [2], and sustainability [3] impacts. About 46.3 million metric tonnes of feed was fed to pigs in the United States in 2016, consisting of 16% corn distillers dried grains with solubles (cDDGS), and 15% total soybean products [4]. However, these ingredients contain higher amounts of DF and less starch compared with corn, resulting in a greater production of short-chain fatty acids (SCFA) by gut microbiota when pigs are fed diets containing cDDGS or soybean hulls (SBH) than corn. The SCFA affect the intestinal epithelial cells and affect the intestinal integrity by regulating ion absorption and gut motility [5].

Various in vitro methodologies have been developed and used to estimate the digestibility of various feed ingredients, including ingredients that contain high concentrations of DF. The most widely used procedure is a three-step in vitro assay that combines replicated enzymatic hydrolysis from the stomach through small intestine [6] with representative large intestine fermentation using swine feces as a living bacterial inoculum [7]. This procedure has been well accepted to estimate in vitro dry matter digestibility (IVDMD) in the large intestine and total gas production of various feed ingredients for swine [5,8,9,10,11]. An automated recording system (ARS) for gas production (GP) was introduced in the early 1990s to reduce the amount of labor, compared with the manual recording system (MRS) when evaluating diets and feed ingredients for ruminants [12]. The ARS technique measures the kinetics of microbial fermentation in an automated fashion by monitoring the gas pressure and ventilation process [12]. Several in vitro studies have investigated the advantages and disadvantages of using ARS to measure the gas production profile and fermentation kinetics in ruminant-based in vitro systems [13,14]. However, the type of feed ingredient, amount of fecal inoculum, quantity of substrate, and the type of recording system may affect the accuracy and precision of the parameters estimated.

Our current study was conducted to determine the effects of inoculum volume and recording system on in vitro gas production and the concentration of SCFA produced from the fermentation of cDDGS and SBH. This investigation was based on the hypothesis that the type of ingredient, volume of fecal inoculum, and amount of substrate in a bottle would affect the accuracy and precision of gas production parameter measurements, including the concentration of SCFA when using the ARS in a pig-based in vitro digestibility system.

## 2. Materials and Methods

### 2.1. Experimental Design, Feed Samples, and Enzymatic Hydrolysis

This experiment was conducted using a 2 × 2 × 2 factorial arrangement of treatments to examine the effects of feed ingredients (cDDGS or SBH), fecal inoculum volume (IV30 = 200 mg substrate + 30 mL inoculum or IV75 = 500 mg substrate + 75 mL inoculum), and GP recording system (MRS or ARS) on IVDMF and the production of SCFA. Hydrolyzed corn DDGS and SBH residues were obtained from the two-step procedure involving pepsin and pancreatin hydrolysis in our previous studies [8,15] that was developed by Boisen and Fernandez [6]. Briefly, 2 g of each cDDGS and SBH sample was weighed into a 500-mL Pyrex Erlenmeyer flask and incubated at 39 °C in a water bath. Then, 100 mL of phosphate buffer solution (0.1 M 7:1 KH_2_PO_4_:Na_2_HPO_4_, pH 6.0) and 40 mL 0.2 M HCl solution (pH 2.0) were added. The pH was adjusted to 2.0 by adding 1 M HCl or 1 M NaOH. The addition of 2 mL of 5 mg/mL chloramphenicol (C0378; Sigma-Aldrich Corp., St. Louis, MO, USA) solution (dissolved in ethanol) was added to prevent bacterial growth during hydrolysis. A volume of 4 mL of 100 mg/mL fresh porcine pepsin (P7000, 421 pepsin units / mg solids; Sigma-Aldrich Corp.) solution (dissolved in 0.2 M HCl) was added to each bottle and incubated in a water bath at 39 °C for 2 h. All the flasks were shaken gently by hand for 5 s every 15 min. Subsequently, 40 mL of 0.2 M phosphate buffer (7:1 KH_2_PO_4_:Na_2_HPO_4_, pH 6.8) and 20 mL of 0.6 M NaOH were added to each flask. Finally, 4 mL of 100 mg/mL fresh porcine pancreatin (P1750, 4 times the specifications of the United States Pharmacopeia; Sigma-Aldrich Corp.) solution (dissolved in 0.2 M phosphate buffer) was added. The hydrolysis continued for 4 h under the same conditions as used for pepsin hydrolysis. Subsequent in vitro fermentation analysis was performed using these residues according to the procedure developed by Jha et al. [10,11].

### 2.2. Experimental Design and In Vitro Fermentation Procedures

Before fermentation, samples of cDDGS and SBH were hydrolyzed by enzymatic digestion with pepsin and pancreatin. The residues from enzymatic digestion were then subsequently pooled within each ingredient source for in vitro fermentation. Blank inocula without substrates were used as controls. The experimental scheme was as follows: 8 treatments × 3 replications + 4 blanks repeated over three batches. Briefly, either 0.2 g or 0.5 g of pooled hydrolyzed cDDGS and SBH samples (ground to 1 mm in particle size) was weighed and incubated in a buffer solution containing macro-and micro-minerals [16]. Feces were collected by rectal stimulation from one finishing pig per batch. Pigs were fed a conventional corn-soybean meal-based diet without antibiotics (Innovation Campus, Cargill Animal Nutrition, Elk River, MN, USA). Collected fecal samples were immediately placed in air-tight plastic syringes and kept in a water bath at 39 °C until incubation. The time from fecal collection until incubation was less than 1 h. In the laboratory, the inoculum was formulated by diluting blended feces in an inoculation solution composed of distilled water (474 mL/L), trace mineral solution (0.12 mL/L containing 132 g/L of CaCl_2_, 100 g/L of MnCl_3_·4H_2_O, 10 g/L of CoCl_2_·6H_2_O, and 80 g/L of FeCl_3_·6H_2_O), in vitro buffer solution (237 mL/L containing 4.0 g/L of NH_4_HCO_3_ and 35 g/L of NaHCO_3_), macromineral solution (237 mL/L composed of 5.7 g/L of Na_2_HPO_4_, 6.2 g/L of KH_2_PO_4_, 0.583 g/L of MgSO_4_·7H_2_O, and 2.22 g/L of NaCl), and resazurin (blue dye, 0.1% wt/vol solution; 1.22 mL/L) and filtered through four layers of cheesecloth. The final inoculum concentration was 0.05 g feces/mL of buffer. Either 30 mL or 75 mL of inoculum aliquots were respectively transferred into bottles containing 200 mg or 500 mg of the hydrolyzed sample substrates to provide an equal inoculum to substrate ratio (6.67 mL/mg) between the two systems. Carbon dioxide (CO_2_) was provided to maintain an anaerobic environment during the entire inoculum preparation process.

The headspace gas pressure in the bottles was recorded using either MRS or ARS. The gas was measured manually at 11 time points post-inoculation using an inverted 25-mL burette with its stopcock end attached to the vacuum, and its open end submerged into a water bath (39 °C) in MRS. The ARS was designed to measure the kinetics of microbial fermentation by monitoring the gas pressure automatically every 5 min and recording remotely using a commercial apparatus (Ankom^RF^ Gas Production System, Ankom Technology, Macedon NY, USA) equipped with real-time sensors. The headspace volume was 57.5 mL in MRS and 257.5 mL in ARS. For the ARS system, accumulated gas in the headspace was automatically released when the pressure exceeded 35 psi. Recording of headspace pressure was terminated at 72 h post-incubation. At the end of the 72 h, the supernatant from each bottle was collected for SCFA analysis.

### 2.3. Chemical Analyses

Before liquid chromatography–mass spectrometry (LC-MS) analysis, samples of fermentation supernatants were derivatized with hydroquinone (HQ) for the determination of SCFA concentrations [17]. Briefly, two microliters of the extracted supernatant were mixed with 70 µL of acetonitrile (ACN) containing 7.5 µM acetic acid-d4, 10 µL dipridyl disulfide (DPDS), 10 µL triphenylphosphine (TPP), and 10 µL HQ. The mixture was incubated at 60 °C for 30 min, chilled on ice, and mixed with 100 µL H_2_O. The vials were then centrifuged at 21,000 × *g* at 4 °C for 10 min. The processed HQ-reaction mixture from chemical derivatization of samples was injected into ultra-performance liquid chromatography (UPLC) system (Xevo-G2-S; Waters, Milford, MA, USA). The concentration of individual compounds was determined by calculating the ratio between the peak area of compounds and the peak area of internal standards. Acetic acid-d4 was used as an internal standard calibration curve for precise SCFA quantification. The acquired data were processed by software (QuanLynx, Waters, Milford, MA, USA).

### 2.4. Calculations

The in vitro fermentability of dry matter (IVDMF) during fecal inoculum fermentation was calculated as follows:IVDMF, % = [(dry weight of the hydrolyzed residue − dry weight of the residue after fermentation)/dry weight of the hydrolyzed residue] × 100

After correction for the blank units, the recorded cumulative gas pressure (psi) was converted into mL of gas produced per g DM using Avogadro’s law as follows:Gas volume, mL = gas pressure × [V/RT] × 22.4 L/mol × 1000 mL/L,
where V denotes head space volume in the bottle (L), R was the gas constant 8.314472 L k Pa/K/mol, and T represents the temperature in Kelvin (273 °K + Celsius temperature in the bottle).

Gas accumulation curves (mL/g DM) recorded during the 72 h of fermentation were fitted by the following model developed by France et al. [18]:
*G* (mL /g DM) = 0, if 0 < *t* < *L*
$$G=G_{f}\;(1-\exp\;(-[b\;(t-L)+c\;(\sqrt{t}-\sqrt{L})])),\;\mathrm{if}\;t\geq L,$$
where *G* denotes the gas accumulation at a specific time (t), *G_f_* (mL/g DM) was the maximum gas volume for *t* = ∞, and *L* (h) represents the lag time before the fermentation began and is determined by the initial delay until the onset of gas production occurs. In the present study, gas accumulation of the cDDGS treatment rapidly reached one-fourth of the maximum accumulation in 2 h, and the parameter *L* (h) was very close to 0, which resulted in the model failing to converge. Therefore, *L* (h) data were removed from the final model. The constants b (h^−1^) and c (h^−1/2^) determine the fractional rate of degradation of the substrate µ (h − 1), which is postulated to vary with time as follows:$$\mu =b+c/(2\sqrt{t}),\;\mathrm{if}\;t\geq L$$

Kinetics parameters of gas production (*G_f_*, *T*/2, G72, and *μ* at T/2) were compared in the statistical analysis, with T/2 representing the time to half asymptote when *G* = *G_f_*/2.

### 2.5. Statistical Analyses

The kinetics of gas production parameters were fitted based on the individual time series data and were analyzed using PROC NLIN of SAS version 9.4 (SAS Inst. Inc., Cary, NC, USA). The IVDMF, fitted gas production kinetic parameters, and the concentration of SCFA were analyzed using the GLIMMIX procedure of SAS version 9.4 (SAS Inst., Inc., Cary, NC, USA), with individual bottles considered as the experimental unit. The model included substrates (cDDGS and SBH), inoculum volume (30 mL and 75 mL), GP recording system (MRS and ARS), and their interactions (Substrate × Volume, Substrate × System, Volume × System, and Substrate × Volume × System) as the fixed factors and batches of samples as random factors. The average coefficient of variance (CV) was calculated based on the average values of kinetic parameters within each treatment using PROC GLM of SAS version 9.4 (SAS Inst., Inc., Cary, NC, USA). The least square means of individual treatments were separated by the Tukey method. Results were considered significant at *p* ≤ 0.05 and trends at 0.05 < *p* ≤ 0.10.

## 3. Results and Discussion

### 3.1. Fermentation Kinetics and Metabolites

Soybean hulls yielded greater (*p* < 0.01) maximal gas production (*G_f_*), required less time (*p* < 0.02) to achieve half gas accumulation (*T*/2), and had greater (*p* < 0.01) fractional rate of degradation (*µ*) and IVDMF compared to cDDGS (Table 1). Each of ingredients showed similar gas production curves regardless of gas recording system and inoculum volume (Figure 1). Results for IVDMF of cDDGS (69.2%) obtained in the current study were greater than that reported in previous studies (59.6% by Jha et al. [9]; 55.7% by Huang et al. [8]), but maximum gas volume (G_f_) of cDDGS (200 mL/g DM) was comparable to those reported by Jha et al. [9] (200 mL/g DM) and Huang et al. [8] (208 mL/g DM). Different kinetics of GP between these two ingredients can be explained by their fiber composition. Soybean hulls contain about 5.5 times more soluble dietary fiber (SDF) than insoluble dietary fiber (IDF), whereas cDDGS contains 1.6 times more SDF than IDF [19]. It has been suggested that apparent ileal digestibility (AID) and apparent total tract digestibility (ATTD) of SDF are a result of greater fermentation compared with IDF in growing-finishing pigs [20]. Moreover, while SDF is mainly fermented in the proximal colon, IDF is fermented primarily in the distal colon [21], which is likely due to the hydrophobic and the crystalline characteristics of these types of DF [22]. Consequently, a greater SDF/IDF ratio in SBH may have resulted in a sharp increase in the fractional rate of degradation during earlier fermentation stage (<8 h) compared with cDDGS in the current study.

Gas production (GP) kinetics parameters were not different between the GP recording systems, whereas IVDMF was less (*p* < 0.01) when recorded in the ARS system compared with the MRS system (79.0% vs. 81.2%, respectively). Moreover, interactions were observed between GP recording system and inoculum volume and substrate quantity for *G_f_* (*p* < 0.04) and *µ* (*p* < 0.01), and the time to half asymptote (*T*/2, *p* < 0.04). According to the meta-analysis on methodological factors influencing GP during in vitro rumen fermentation, the GP recording apparatus with venting system (i.e., ARS) resulted in greater gas production estimates compared to the MRS GP recording apparatus operating without venting system [23]. Furthermore, the absence of automatic ventilation system in MRS increased headspace pressure, so that it may have caused a partial dissolution of carbon dioxide (CO_2_) in the inoculum, and subsequently resulted in the underestimation of GP as well as restricting microbial respiration. Results from the current experiment showed no difference in parameters of GP kinetics between the two systems. One possible explanation for the lack of differences may be due to the differences between the headspace volumes to fermentation inoculum ratio between the systems, which was 4.9 for ARS compared to 2.8 for MRS in the current experiment. This ARS ratio is greater than the ratio used in a previous in vitro study conducted using swine fecal inoculum with ARS (ratio: 3.2, Pastorelli et al. [24]). However, the optimal ratio between headspace and fermentation inoculum has not yet been established. The smaller ratio may result in greater underestimation of GP because of higher pressure [25], whereas the larger ratio may result in lower pressure and cause inhibition of microbial activity [13]. Therefore, based on the current results, it can be expected that relatively larger ratio between headspace volume to inoculum in ARS may interfere with the microbial fermentation in the bottle, resulting in decreased IVDMF, as well as increased within batch variation. However, further investigations are required to determine the optimal ratio between headspace and inoculum volume when using swine fecal inoculum in ARS.

Regardless of substrates, acetic acid was the most abundant SCFA produced during in vitro fermentation. The samples of SBH produced more acetic acid (*p* < 0.01), propionic acid (*p* < 0.05), and total SCFA (*p* < 0.01) compared with cDDGS (Table 1). These results are in agreement with those from a previous in vitro study by Jha and Leterme [26], indicating that both in vitro GP recording systems yielded accurate estimates of microbial fermentation. The greater SCFA production observed during fermentation of SBH compared with cDDGS can be attributed to the solubility of DF in the ingredient. Ferulic acid consists of cross-linked cell wall polysaccharides and other cell wall components such that it might be associated with fiber matrix rigidity [27]. Insoluble DF has been linked to decreased SCFA production resulting from slower fermentation rates compared to soluble DF, and insoluble DF contains 100 times greater ferulic acid content than soluble DF [28]. Thus, there was a greater amount of soluble dietary fiber available for microbiota fermentation in SBH, resulting in increased production of SCFA compared to cDDGS (Figure 1).

### 3.2. The Average Coefficient of Variance

The hypothesis of this study was that error frequency and severity would be relatively greater in MRS compared to ARS because of the intensive labor involved during the first phase of microbial fermentation. Although we observed no differences between the two GP recording systems for the coefficient of variation (CV) of GP kinetic parameters and IVDMF, the CV tended to be less (*p* < 0.10) when using the greater inoculum volume (IV75) compared to using the smaller inoculum volume (IV30, Table 2). Also, the CV tended to be less in cDDGS on time to half asymptote (*T*/2, *p* < 0.07) and IVDMF (*p* < 0.09) compared to SBH.

Comparison of results from our variability analysis to results from other studies is difficult because each study analyzed results using different mathematical methods. However, results from a previous study evaluating the repeatability and reproducibility of an ARS using rumen fluid from four laboratories indicated that fermentable organic matter had the greatest repeatability and reproducibility (0.2 to 1.9%, and 0.3 to 4.5%, respectively), followed by kinetic parameters (*Gf* = 1.1 to 2.5% in repeatability and 1.7 to 3.8% in reproducibility; *T*/2 = 4.3 to 13.2% in repeatability and 4.7 to 13.2% in reproducibility; *µ*= 8.2 to 12.8% in repeatability and 18.6 to 27.5% in reproducibility) [29]. This pattern was similar to the results obtained in the current study, indicating that the CV for IVDMF had the least variation, and kinetic parameters showed comparatively greater variation. Also, a similar CV pattern of kinetics was observed in a recent study using the ring test of in vitro GP recording systems conducted in four different laboratories in Europe (Denmark, United Kingdom, Spain, Italy), using the same wireless apparatus that we used in the current experiment [30]. These researchers also indicated that the least variation among parameters of GP kinetics observed were as follows: GP at 48 h (CV = 4.8%), *Gf* (CV = 6.4%), *µ* (CV = 11.4%), and *T*/2 (CV = 14.1%).

Rymer et al. [31] indicated that the largest source of variation in the GP technique could be attributed to the source of inoculum and its microbial activity. In our study, we assumed that the fecal samples may vary among fecal donor age and may significantly increase the CV in the current experiment. Fecal sampling procedures were irregularly managed because of the bio-security of the company-owned research farm. There is a relatively large variation in age and body weight (60 to 100 kg) of fecal donors between batches. Kim et al. [32] indicated that pig microbial ecosystems in the GIT continue to change as pigs grow, and is influenced by various factors, including genetics, diet, and antibiotics. Therefore, our results reflect the fact that using fecal inocula from pigs of different ages leads to differences in microbial fermentability derived from different microbial communities within the batch of samples, resulting in increased variability of GP kinetic curves. The use of inoculum from one fecal donor per batch can be another factor. In our previous in vitro study, fecal samples were randomly collected from three out of five growing pigs for each batch of feed ingredient samples analyzed [8], resulting in CV’s of kinetic parameters (*Gf*, and *T*/2) of 5.2 and 4.5% in SBH, respectively, and 9.8 and 18.5% in cDDGS, respectively, which were 5.18 and 2.81 times less than the CV’s obtained from the current experiment. Rymer et al. [31] emphasized that fecal samples should be collected from several animals for in vitro fermentation analysis because each pig has different fecal microflora composition even though they are from the same genetic line and consume the same diet. Evidence from human studies has shown that using inoculum from at least three donors may enhance the predictive value of in vitro colonic fermentation [33]. Based on the results from the current experiment, we suggest collecting fecal samples from more than three pigs is necessary for improving the accuracy of pig in vitro fermentation assays of high fiber ingredients.

## 4. Conclusions

The results of this experiment demonstrate that both the GP recording systems (manual and automatic) were accurate at recording the gas production during in vitro fermentation similar to results reported in the literature for cDDGS and SBH. These results also suggest that there is an improvement in precision when larger volumes of fecal inocula are used if the ratio of substrate and headspace are kept in proportion between the GP recording systems. 

## Figures and Tables

**Figure 1 animals-09-00773-f001:**
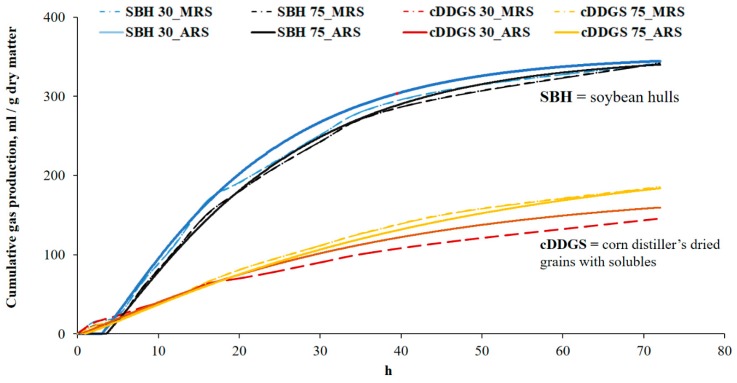
Gas accumulation curves of two ingredients (soybean hulls = SBH; and corn dried distiller’s grains with solubles = DDGS) and inoculum volume (30 and 75 mL) incubated either in automatic gas production recording system (ARS) or manual gas production recording system (MRS) during 72 h.

**Table 1 animals-09-00773-t001:** Parameters of the fitted kinetics and concentration of short-chain fatty acids (SCFA) after in vitro fermentation of corn distillers dried grains with solubles (cDDGS) and soybean hulls (SBH) using an automatic recording system (ARS) and manual recording system (MRS) using different fecal inoculum volumes (30 and 75 mL) obtained from pigs ^1^.

Item	ARS		MRS		SEM ^2^	Substrates		SEM	*p*-Values ^3^			
	30 mL	75 mL	30 mL	75 mL		cDDGS	SBH		Sub	Sys	Vol	Sys × Vol
	Fermentation kinetics											
N ^4^	8	8	8	8		20	20					
Gf ^5^	332 ^a^	256 ^b^	281 ^ab^	286 ^ab^	37.4	200 ^B^	362 ^A^	34.3	0.011	0.971	0.334	0.047
µ ^6^	0.04 ^b^	0.05 ^a^	0.04 ^ab^	0.04 ^ab^	0.015	0.02 ^B^	0.05 ^A^	0.014	0.012	0.446	0.544	0.012
T/2 ^7^	34.34 ^a^	25.07 ^b^	27.16 ^ab^	28.16 ^ab^	7.832	32.02 ^A^	26.10 ^B^	7.794	0.028	0.984	0.977	0.041
G72 ^8^	211	210	224	240	12.2	159 ^B^	285 ^A^	9.8	0.016	0.987	0.826	0.110
IVDMF ^9^	79.2	79.0	80.8	81.0	1.10	69.2 ^B^	90.9 ^A^	0.93	0.012	0.012	0.731	0.463
	Fermentation metabolites, mmol/g											
Acetic acid	4.78	4.19	4.32	4.53	0.504	3.71 ^B^	5.20 ^A^	0.504	0.009	0.906	0.346	0.429
Propionic acid	1.14	1.17	1.10	1.12	0.063	1.08 ^B^	1.18 ^A^	0.031	0.047	0.284	0.433	0.917
Butyric acid	0.58	0.71	0.54	0.58	0.055	0.64	0.57	0.054	0.175	0.130	0.271	0.414
Valeric acid	0.19	0.16	0.15	0.22	0.034	0.19	0.16	0.024	0.306	0.688	0.570	0.108
Total SCFA	6.69	6.23	6.10	6.46	0.584	5.64 ^B^	7.11 ^A^	0.413	0.001	0.757	0.428	0.485

^A,B^ Means within a row with different uppercase superscript letters differ significantly (main effect, *p* < 0.05). ^a,b^ Means within a row with different lowercase superscript letters differ significantly (Sys × Vol, *p* < 0.05). ^1^ The least squares mean value presented based on the three replications per treatment. ^2^ Standard error of the means. ^3^ Sub = feed ingredients as substrates; Sys = gas production recording system; Vol = inoculum volume. ^4^ Number of observations in fermentation. ^5^ Maximum gas volume (mL/g DM incubated). ^6^ Fractional rate of degradation (h − 1) at t = *T*/2. ^7^
*T*/2, half-time to asymptote (h). ^8^ Volume of gas production at 72 h. ^9^ In vitro dry matter fermentability.

**Table 2 animals-09-00773-t002:** Average coefficient of variation of the in vitro fermentation kinetic parameters for corn distillers dried grains with solubles (cDDGS) and soybean hulls (SBH) using an automatic recording system (ARS) and manual recording system (MRS) with two different pig fecal inoculum volumes (30 and 75 mL) ^1^.

Item	ARS		MRS		SEM ^2^	Substrates		SEM	*p*-Values ^3^			
	30 mL	75 mL	30 mL	75 mL		cDDGS	SBH		Sub	Sys	Vol	Sys × Vol
Gf ^4^	29.2	23.7	31.5	13.9	12.53	22.7	26.4	8.16	0.757	0.758	0.100	0.762
µ ^5^	43.7	45.7	50.6	38.5	14.49	31.2	58.0	6.06	0.068	0.985	0.688	0.941
T/2 ^6^	32.2	48.9	55.6	27.0	16.06	29.1	52.8	9.20	0.122	0.958	0.716	0.590
G72 ^7^	33.6	20.9	10.2	9.8	5.53	19.9	17.2	5.92	0.750	0.136	0.447	0.203
IVDMF ^8^	4.35	3.15	3.85	4.85	0.03	5.02	3.08	0.81	0.121	0.588	0.929	0.764

^1^ The least squares mean value presented based on the three replications per treatment. ^2^ Standard error of the means. ^3^ Sub = feed ingredients as substrates; Sys = gas production recording system; Vol = inoculum volume. ^4^ Maximum gas volume (mL/g DM incubated). ^5^ Fractional rate of degradation (h − 1) at t = *T*/2. ^6^
*T*/2, half-time to asymptote (h). ^7^ Termination gas volume (at 72 h). ^8^ In vitro dry matter fermentability.

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
