# Peer review of "Effects of Gas Production Recording System and Pig Fecal Inoculum Volume on Kinetics and Variation of In Vitro Fermentation using Corn Distiller’s Dried Grains with Solubles and Soybean Hulls"

_animals, 2019, doi:10.3390/ani9100773_

Round 1

Reviewer 1 Report

Volume of inoculum and quantity of substrate are confounded. This should be clearly acknowledged throughout.

Additional suggested changes are in the attachment.

Author Response

   This study detected three factors with two levels of each factor, actually significant difference was mainly found between substrates. I recommend the authors should mainly focus on one factor and design more levels to evaluate the true variation. Regarding the fecal inoculum for in vitro system, a series of different amounts of substrate, and inocula can be tested. In addition, the authors can also focus on comparing the fermentation characteristic between cDDGS and SBH, but more parameters should be detected.

   Unlike ruminant in-vitro studies, in-vitro fermentation studies using pig fecal inoculum with automatic recording system is limited. Therefore, we set up factorial (2 x 2 x 2) design to investigate the correlation between fecal inoculum volume and gas recording system on the fermentation kinetics curve, variance, and SCFA production. Levels on the inoculum volume were based on previous in-vitro studies (both ARS and MRS), which makes our studies relevant. Due to the headspace pressure, the inclusion of inoculum more than 75mL is not recommended according to the instructions of the manufacturer.

   This study is the consecutive work based on our previous studies (Saqui-Salces et al., 2016; Huang et al., 2017), and the concept of this study is to provide knowledge (scientific evidence) on the fermentation characteristics of two fiber ingredients. Reason for the use of the kinetics parameters measured in this study is for compatibility. In doing so, the authors can easily compare the value of fermentation kinetics with other literature and estimate expected SCFA production as well as reproducibility. We do not think that more parameters are needed for this study.

Reviewer 2 Report

The authors determined the effects of gas production recording system and pig fecal inoculum volume on kinetics and variation of in vitro fermentation. The manuscript is well written and easy to read. I have only very minor comments on this manuscript.

Comments

L 25: Please be consistent in the unit for the amounts of substrate. Here, the authors used “g” but, in the M&M section “mg.”

L 53: Please do not begin a sentence with an abbreviation here and throughout the manuscript.

L 74: Please change “short chain fatty acids (SCFA)” to “SCFA” as the author already defined the SCFA in L 52.

L 75: Please italicize “in vitro”

L 79: Change “factorial design” to “factorial treatment arrangements”

L 81: Change “(MRS = manual recording system or ARS = automated recording system)” to “(MRS or ARS)”as these abbr. have been defined in L 62 and 63; Please double check “IVDMF” would this be “IVFDM”?? was this defined previously?

L 87: Please change “ml” to “mL” Please be consistent here and throughout the manuscript.

L 92: Please add a space to make “39 °C”

L 97: Please italicize “in vitro”; Delete “-” to make “in vitro”

L 107: … soybean meal-based …

L 113: Please italicize “in vitro” here and throughout the manuscript.

L 151: Please add a comma to make “1,000 mL”

L 185: “IVFDM” has been readily defined; Add a table number to make “…with cDDGS (Table 1).”

L 200: Please double check the value of IVFDM in ARS and MRS systems. The values in the text are not supported by the data in Table 1.

L 230: Please double check “diferulate content” a typo??

L 267: Please double check defined word “IVFDM” organic matter or what?

Table 1:

Here and in other Tables, please change “ml” to “mL”

Please present 3 places of decimals for P-values of treatments

Delete “(LS)”

in-vitro fermentation of dry matter > in vitro fermentability of dry matter

Table 2:

Delete “(CV)”

Please present 3 places of decimals for P-values of treatments

Delete “(LS)”

in-vitro fermentation of dry matter > in vitro fermentability of dry matter

Figure 1.

Please define “soybean hulls” and “corn dried distiller’s grains with solubles”

The authors may be interested in referring the following paper: Choi, H., J. Y. Sung, and B. G. Kim. 2019. Neutral detergent fiber rather than other dietary fiber types as an independent variable increases the accuracy of prediction equation for digestible energy value in feeds for pigs. Asian-Australasian Journal of Animal Sciences doi: 10.5713/ajas.19.0103

Author Response

L2: Suggestion: You are optimizing how to get the lowest variance with the lowest workload, yes? Maybe you can let the title reflect that.

I amended the titles (L2).

L12 & L19: I propose you add a small intro as to why this is relevant. It is described in the introduction, so please a sentence or so for the reader to ease into the text.

I included an introductory sentence in simple summary section (L12-13).

L26: recoding should be recording

I changed.

L45: what exactly is the reasoning for the study in a practical sense, e.g. why these operating parameters? Is it because of experiment throughput or decreased variation as hinted in the abstract?

In a practical sense, ARS gives more convenience compared with MRS, fundamentally ARS reduced night shift as well as labor cost compared with MRS. This study was investigated to evaluate reproducibility between ARS and MRS with modification of inoculum volume and substrates. The fermentation kinetic parameters measured in our study was very essential and common (well-known) units to estimate the amount of gas production as well as SCFA production.

L51: gas production is a bacterial byproduct, giving very little energy to the animal. Rather SCFA is the main contributor here, and I see little reason for this to be highly correlated. As it stands now, the reason for investigating gas production is to investigate the metabolizable energy for pig, which I suggest you either add more evidence for or rewrite.

I rewrote the sentence (L50-55).

L57: Sounds more like the digestibility of the upper GI. Can you add data, literature or estimates on how much of this would be different in an actual animal, e.g. where most of the pepsin/pancreatin-released matter would be absorbed in the jejunum. Presumably this would account for much of the initial rates of the fermentations.

I do not find appropriate literature on it. No studies noted the value of the absorption in the jejunum regarding pepsin/pancreatin enzyme secretion. I suspect that the amount of enzymes secretion vary depending on what ingredients were digested in the pig model. I think your suggestion more fits the in-vivo hindgut digestibility trial.

L64: do you disagree with their conclusions? What specifically has been left out and given the wrong data in your estimation? How exactly would an automated system theoretically interact with the other operating variables? I imagine a major difference is the release of pressure in the ARS system?

I agree with your opinion that the headspace pressure makes a major difference in ARS on the kinetic parameters in a theoretical manner. Although the current experiment did not show differences between gas recording system and inoculum volume, further research is required. Possible modification can be found by the study of Tagliapieta et al. (2010), which I cited. The authors noted that stirring during fermentation is not needed in ARS, which is not commonly accepted in the in-vitro analysis using pig fecal inoculum. To provide a peristaltic wave in the large intestine, bottles should be stirred by the end of the fermentation.

L67: a hypothesis is a proposed explanation for a phenomena or an educated guess, and you haven’t actually provided a theoretical reasoning as to why this could be the case apart from that it may be so in L65. Please elaborate on the theoreticals here.

I amended the paragraph (L70-75).

L79 & L81: please elaborate on these procedures rather than refer, the details are for interpretation.

I added the procedures (L85-97).

L90: How many animals and how was the fecal matter distributed? The fecal microbiome can differ immensely among animals, which is why pooling and homogenization of multiple samples is usually preferred – I wonder if this was done here when you say blended? Please elaborate on how you did your batches etc (it is described somewhat in the discussion)

I corrected the sentence (L107).

L95: Please state the fecal concentration and to what degree particulate matter was removed.

I added the procedures (L111-117).

L97: wrong unit, should be mL/mg

Changed

L99: which reducing agents?

Amended (L119-120).

L110: what precisely is meant by washing?

It means “washed by DI water”. I deleted the sentence to reduce confusion.

L137 & L137: I suggest that units are taken out of the equation and rather is described in the text

I changed (L155).

L140: how is lag time determined?

Lag time refers to the time period before fermentation started, and is determined to the initial delay in the onset of gas production (h). I additionally explained the sentence.

L144: So what is the actual used model? I can see in Figure 1 that there is a lag-phase for some graphs

I rewrote the sentence (L162).

L149: Elaborate that the models were fitted on individual time series, and the set of coefficients then were extracted (right?)

Amended (L170).

L156: An F-test would be the appropriate test for equality in variance if that is what you are interested in. I don’t understand how you compare them either, since you necessarily only have one value for each treatment?

I amended the statistical methods of the CV analysis. As you acknowledged, GLM procedures in SAS included F-test, and there was no significance on the F-test results in our data, indicating variance is equally distributed between treatments. Consequently, I do not want to present this F-value as any of the other in-vitro studies neither presented.

L179: please discuss the reasons for the inoculum effect in DDGS.

I do not clearly understand your suggestion. What do you mean by the ‘inoculum effect in DDGS’? I think that I already give some possible suggestions at the very last part of the paragraph (L216-220).

L186: did you measure pH? Presumably carbon dioxide would be buffered by the short chain fatty acids?

To instant liquid collection (for SCFA analysis) at the end of the fermentation stage, I did not measure the pH. Studies Pastorelli et al. (2014) indicated that SCFA can be a contributing factor in lowering pH and gas production, but none of the studies suggested the buffer effect of carbon dioxide by SCFA, presumably indicating that the effect would be minor and/or negligible.     

L214: reorder DDGS75_MRS and DDGS30_MRS for consistency

I reordered. 

L236: I wonder if this section will be changed by the use of the F-test as well.

As I mentioned previously, F-test is already conducted.

L220 & L280: I am finding the table difficult to read, the main reason being that you are trying to show a 2x2x2 design in a 2D table. Presumably, each substrate should be a third level sub-category underneath the volumes. For example, which substrate is used in ARS, 30mL? Which volume and system is used for cDDGS?

I do not agree with your comment. Reading the table is pretty easy as I divided into ‘main effect’ and ‘interaction’ on the footprints respectively. The major purpose of this study was to investigate the interaction between newly introduced gas recording system (ARS) and inoculum volume, not comparing kinetic and SCFA value between substrates. Presenting a sub-category and show 8 treatments in a table will make readers more confused.

Reviewer 3 Report

Comments, review of manuscript id: Animals-608394

Title: ”Effects of gas production recording system and pig fecal inoculum volume on kinetics and variation of in vitro fermentation using corn distillers droed grains with solubles (cDDGS) and soy bean hulls (SBH)”

This manuscript presents the evaluation of two in vitro gas production systems when using two different fecal inoculum volumes. The manuscript is in general well written, and the experimental design and the results is well described. The English language is in general good, but some places, as in lines 281 – 283, page 8, it sounds Chinese-English for me. Please check this sentence. I have no further comments to the manuscript, except for a comment in line 14, page 1. Here you are mention “CP” for the first time. The abbreviation CP is defined later in the manuscript, but it should be written in full as "dietary fiber" in the simple summary.

I also wonder why some parts of the manuscript is written in red.

Author Response

Line 50 - insert – dietary gross energy

I deleted the sentence based on the suggestion of reviewer 2.

Line 53 - change - various to numerous

I deleted the sentence based on the suggestion of reviewer 2.

Line 70 - delete - Therefore, start with Our

Amended (L70).

Line 82 - change - using to according

Changed (L 98).

Line 91 - How many pigs were collected?

Changed (L 106).

Line 264 - change - to to and

Changed (L 284).

This manuscript is a resubmission of an earlier submission. The following is a list of the peer review reports and author responses from that submission.

Round 1

Reviewer 1 Report

This study evaluated two in vitro gas production recording systems (manual vs automated) and the initial fecal inoculum volume (30 vs. 75 mL) on parameters of in vitro fermentation of corn distillers dried grains with solubles (cDDGS) and soybean hulls (SBH). The results showed that the use of 75 mL inoculum volume tended to reduce the variation of measurements compared with 30 mL inoculum volume regardless of gas production recording system. These findings suggest that using larger inoculum volume increases the precision of measurements while the automated system decreases labor for conducting the assay. 

However, this study detected three factors with two levels of each factor, actually significant difference was mainly found between substrates. I recommend the authors should mainly focus on one factor and design more levels to evaluate the true variation. Regarding the fecal inoculum for in vitro system, a series of different amounts of substrate, and inocula can be tested. In addition, the authors can also focus on comparing the fermentation characteristic between cDDGS and SBH, but more parameters should be detected.

Reviewer 2 Report

The authors describe an investigation of gas production parameters across inoculum volume, measurement method and fermentation substrate. The paper is fairly clearly written, but would benefit from some corrections such as some more details in the introduction, mainly a clear description of why this research moves the field forward, which I know is there, but needs more specification. The methods are sound and well thought out, but requires some expansion.

I am curious as to how exactly the comparison of CV% where done, and propose looking into using an F-test instead or as an adjunct.

I have suggested some specifics below.

L2: Suggestion: You are optimizing how to get the lowest variance with the lowest workload, yes? Maybe you can let the title reflect that.

L12 & L19: I propose you add a small intro as to why this is relevant. It is described in the introduction, so please a sentence or so for the reader to ease into the text.

L26: recoding should be recording

L45: what exactly is the reasoning for the study in a practical sense, e.g. why these operating parameters? Is it because of experiment throughput or decreased variation as hinted in the abstract?

L51: gas production is a bacterial byproduct, giving very little energy to the animal. Rather SCFA is the main contributor here, and I see little reason for this to be highly correlated. As it stands now, the reason for investigating gas production is to investigate the metabolizable energy for pig, which I suggest you either add more evidence for or rewrite.

L57: Sounds more like the digestibility of the upper GI. Can you add data, literature or estimates on how much of this would be different in an actual animal, e.g. where most of the pepsin/panceatin-released matter would be absorbed in the jejenum. Presumably this would account for much of the initial rates of the fermentations

L64: do you disagree with their conclusions? What specifically has been left out and given the wrong data in your estimation? How exactly would an automated system theoretically interact with the other operating variables? I imagine a major difference is the release of pressure in the ARS system?

L67: a hypothesis is a proposed explanation for a phenomena or an educated guess, and you haven’t actually provided a theoretical reasoning as to why this could be the case apart from that it may be so in L65. Please elaborate on the theoreticals here.

L79 & L81: please elaborate on these procedures rather than refer, the details are crucial for interpretation.

L90: How many animals and how was the fecal matter distributed? The fecal microbiome can differ immensely among animals, which is why pooling and homogenization of multiple samples is usually preferred – I wonder if this was done here when you say blended? Please elaborate on how you did your batches etc (it is described somewhat in the discussion)

L95: Please state the fecal concentration and to what degree particulate matter was removed.

L97: wrong unit, should be mL/mg

L99: which reducing agents?

L110: what precisely is meant by washing?

L137 & L137: I suggest that units are taken out of the equation and rather is described in the text

L140: how is lag time determined?

L144: So what is the actual used model? I can see in figure 1 that there is a lag-phase for some graphs

L149: Elaborate that the models were fitted on individual timeseries, and the set of coefficients then were extracted (right?)

L156: An F-test would be the appropriate test for equalness in variance if that is what you are interested in. I don’t understand how you compare them either, since you necessarily only have one value for each treatment?

L179: please discuss the reasons for the inoculum effect in DDGS.

L186: did you measure pH? Presumably carbon dioxide would be buffered by the short chain fatty acids?

L214: reorder DDGS75_MRS and DDGS30_MRS for consistency

L236: I wonder if this section will be changed by the use of the F-test as well.

L220 & L280: I am finding the table difficult to read, the main reason being that you are trying to show a 2x2x2 design in a 2D table. Presumably, each substrate should be a third level sub-category underneath the volumes. For example, which substrate is used in ARS, 30mL? Which volume and system is used for cDDGS?

Reviewer 3 Report

Line 50 - insert - dietary gross energy

Line 53 - change - various to numerous

Line 70 - delete - Therefore, start with Our

Line 82 - change - using to according

Line 91 - How many pigs were collected?

Line 264 - change - to to and